# Does Acute Caffeine Supplementation Improve Physical Performance in Female Team-Sport Athletes? Evidence from a Systematic Review and Meta-Analysis

**DOI:** 10.3390/nu13103663

**Published:** 2021-10-19

**Authors:** Alejandro Gomez-Bruton, Jorge Marin-Puyalto, Borja Muñiz-Pardos, Angel Matute-Llorente, Juan Del Coso, Alba Gomez-Cabello, German Vicente-Rodriguez, Jose A. Casajus, Gabriel Lozano-Berges

**Affiliations:** 1GENUD (Growth, Exercise, Nutrition and Development) Research Group, FIMS Collaborating Center of Sports Medicine, 50012 Zaragoza, Spain; jmarinp@unizar.es (J.M.-P.); bmuniz@unizar.es (B.M.-P.); amatute@unizar.es (A.M.-L.); agomez@unizar.es (A.G.-C.); gervicen@unizar.es (G.V.-R.); joseant@unizar.es (J.A.C.); glozano@unizar.es (G.L.-B.); 2Department of Physiatry and Nursing, Faculty of Health and Sport Sciences (FCSD), University of Zaragoza, 22001 Huesca, Spain; 3Physiopathology of Obesity and Nutrition Networking Biomedical Research Center (CIBERObn), 28029 Madrid, Spain; 4Department of Physiatry and Nursing, Faculty of Health Sciences, University of Zaragoza, 50012 Zaragoza, Spain; 5Centre for Sports Studies, Rey Juan Carlos University, 28943 Fuenlabrada, Spain; juan.delcoso@urjc.es; 6Centro Universitario de la Defensa, University of Zaragoza, 50090 Zaragoza, Spain; 7Instituto Agroalimentario de Aragon IA2 (CITA-Universidad de Zaragoza), 50009 Zaragoza, Spain; 8Department of Physiatry and Nursing, Faculty of Medicine, University of Zaragoza, 50012 Zaragoza, Spain

**Keywords:** soccer, volleyball, basketball, ergogenic aid, elite athletes, sports performance

## Abstract

Introduction: Recent original research and meta-analyses suggest that acute caffeine supplementation improves exercise performance in team-sport athletes (TSA). Nonetheless, most of the studies testing the effects of caffeine on TSA included samples of male athletes, and there is no meta-analysis of the performance-enhancing effects of caffeine on female TSA. The aim of the present study was to synthesize the existing literature regarding the effect of caffeine supplementation on physical performance in adult female TSA. Methods: A search was performed in Pubmed/Medline, SPORTDiscus and Scopus. The search was performed from the inception of indexing until 1 September 2021. Crossover randomized controlled trials (RCT) assessing the effects of oral caffeine intake on several aspects of performance in female TSA were selected. The methodological quality and risk of bias were assessed for individual studies using the Physiotherapy Evidence Database scale (PEDro) and the RoB 2 tool. A random-effects meta-analysis of standardized mean differences (SMD) was performed for several performance variables. Results: The search retrieved 18 articles that fulfilled the inclusion/exclusion criteria. Overall, most of the studies were of excellent quality with a low risk of bias. The meta-analysis results showed that caffeine increased performance in specific team-sport skills (SMD: 0.384, 95% confidence interval (CI): 0.077–0.691), countermovement jump (SMD: 0.208, CI: 0.079–0.337), total body impacts (SMD: 0.488; 95% CI: 0.050, 0.927) and handgrip strength (SMD: 0.395, CI: 0.126–0.665). No effects were found on the ratings of perceived exertion, squat jumps, agility, repeated sprint ability or agility tests performed after fatigue. Conclusions: The results of the meta-analysis revealed that acute caffeine intake was effective in increasing some aspects of team-sports performance in women athletes. Hence, caffeine could be considered as a supplementation strategy for female athletes competing in team sports.

## 1. Introduction

The use of caffeine in sporting events was controlled until 1 January 2004, since a post-competition urinary concentration above 12 micrograms per milliliter was considered an adverse analytical finding by the World Anti-Doping Agency [1]. However, at that date, caffeine was removed from the list of prohibited substances in the monitoring program of the World Anti-Doping Agency [2]. The removal of caffeine from the list of prohibited substances, in addition to increasing scientific knowledge about the potential ergogenic effects of caffeine, has caused an increase in caffeine intake in both men and women athletes over recent years [3].

The widespread use of this supplement in sport is based on scientific evidence, as it has been classified by the International Society of Sports Nutrition (ISSN) as a “*Strong evidence to support efficacy and apparently safe*” supplement [4], with recommended doses ranging from 3 to 6 mg per kg of body mass with a timing of ingestion of 1 h before exercise. A vast amount of research indicates that caffeine intake can have a positive effect on several forms of athletic performance [5,6]. Grgic and colleagues performed an umbrella review in 2019 including 21 published meta-analyses, revealing that caffeine supplementation elicited an ergogenic effect on muscle endurance and strength, anaerobic power and aerobic endurance, which are critical variables for team-sports performance [5].

In team-sport athletes (TSA), the efficacy of caffeine supplementation in enhancing performance is less clear than in other sport disciplines, because success is explained by a combination of physical, technical and tactical skills. Brown et al. suggested via a meta-analysis that caffeine had no effect on repeated sprint ability (RSA) in TSA [7]. These results were contradicted by a review performed by Chia et al. [8], who found improvements in sprint performance (in 8 out of 10 studies) and vertical jump (in 7 out of 8 studies) in ball game athletes. These findings were reaffirmed by a later meta-analysis developed by Salinero et al. [9] evaluating TSA and also by a systematic review developed by Mielgo-Ayuso et al. [10] focusing on soccer players. Both studies concluded that acute caffeine ingestion improved jump height and RSA [9,10] in addition to agility performance, total running distance and number of performed sprints during a match [9]. Nonetheless, Ferreira and colleagues [11] recently performed a meta-analysis focusing on the effects of caffeine on soccer, finding no significant improvements in soccer-related performance following caffeine supplementation. Therefore, although the positive effects that caffeine supplementation may have on athletic performance in certain individual sports (e.g., running, cycling, etc.) are evident, it seems that more research is needed to determine the ergogenic effect of acute caffeine intake in team sports.

Moreover, most of the studies included in the above-mentioned systematic reviews and meta-analyses only included male athletes, as stated in a recent letter to the editor by Salinero et al. entitled “*More research is necessary to establish the ergogenic effect of caffeine in female athletes*” [12]. In this letter, the authors analyzed the percentage of females in studies evaluating the ergogenic effects of caffeine, reporting that only 13% of the participants were female. Moreover, although some studies included both male and female participants (contributing to the aforementioned 13%), most of them drew conclusions for the whole sample, irrespective of potential sex differences [13,14].

Despite the lack of research specifically analyzing female athletes, current guidelines for caffeine supplementation are identically applied for both males and females [15]. However, these guidelines were established primarily from studies developed in males, which is a clear limitation and raises concerns about their practicality. Although recent evidence suggests that the pharmacokinetics of acute caffeine intake seems to be similar in all phases of the menstrual cycle and that women athletes benefit from caffeine intake across all phases of the menstrual cycle [16], it is still possible that women obtain lower ergogenic effects of oral caffeine intake due to the interaction of caffeine and female sex hormones [17]. Along these lines, Temple and Ziegler [18] found sex differences in subjective and physiological responses to caffeine that were mediated by changes in circulating steroid hormones. In fact, inconsistent results have been found when comparing the ergogenic effects of caffeine in both sexes, with some studies finding some differences [19] while others found none [20,21]. Moreover, some researchers have concluded that the ergogenic effect of oral caffeine intake is present in both sexes but differs in its magnitude [22]. Along these lines, Mielgo-Ayuso et al. [10] recently developed a systematic review including 10 studies that evaluated the ergogenic effect of caffeine on both males and females. These authors concluded that caffeine supplementation produced a similar ergogenic benefit regarding aerobic performance and fatigue index in men and women, finding larger effects of caffeine intake in men when anaerobic performance was evaluated, which could be critical for team sports. However, the above-mentioned review only focused on the sex comparison; consequently, many studies that only recruited females were excluded, and due to the low number of included studies the authors chose not to perform a meta-analysis.

Thus, the aim of the present study is to perform a qualitative and quantitative analysis of the existing literature regarding the effect of caffeine supplementation on physical performance in adult female TSA.

## 2. Methods

### 2.1. Search Strategy

This systematic review and meta-analysis was carried out following the Preferred Reporting Items for Systematic reviews and Meta-Analyses (PRISMA) 2020 guidelines [23] and was pre-registered in PROSPERO (CRD42021223046). A systematic search was performed in the Pubmed/Medline, SPORTDiscus and Scopus databases. The search was performed from the inception of indexing until 1 September 2021, using the same search syntax as Salinero et al. [9] for Pubmed. An analogous search was performed for SPORTDiscus and Scopus (Appendix A). All articles were downloaded to a CSV document to identify duplicates, and the whole process (i.e., identification, screening and selection of studies) was independently performed by two authors, with any disagreements resolved through discussion.

### 2.2. Inclusion and Exclusion Criteria

The following inclusion criteria were applied to selected studies: (1) studies evaluating the effect of an acute dose of isolated caffeine (e.g., not mixed with other supplements) on physical performance in female TSA (if studies included both sexes we only selected data for females, and if these data were not available we contacted the corresponding author and requested them); (2) studies including adults (18 years of age or over); (3) crossover studies that compared the intake of caffeine and a placebo; (4) studies using a blinded and randomized design; (5) studies in English or Spanish. Studies that supplied doses below 1 mg/kg or above 9 mg/kg, that did not present a true placebo condition (thus not allowing for blinding) or that did not evaluate performance-related variables (e.g., only evaluated oxidative stress markers) were excluded. Note that performance-related variables are explained in detail in the “Data Extraction” section.

### 2.3. Quality Assessment and Risk of Bias

The Physiotherapy Evidence Database scale (PEDro) was used to evaluate the individual quality of each study, with studies being classified as excellent (score 9–10), good (score 6–8), fair (score 4–5) or poor (score < 4). The PEDro scale has been shown to be valid and reliable for assessing the internal validity of randomized controlled trials [24].

Following the Cochrane Collaboration guidelines, the RoB 2 tool for randomized crossover designs was applied to assess the risk of bias of each study included [25]. RoB 2 includes the following domains for crossover trials: (1) bias arising from the randomization process; (2) bias due to deviations from the intended intervention; (3) bias due to missing outcome data; (4) bias in the measurement of the outcome; (5) bias in the selection of the reported results. Due to the characteristics of the crossover design, another domain related to bias arising from the period effect and the carryover effect should be considered (domain S). Finally, each study was classified as having a high risk of bias, some concerns or a low risk of bias.

Both the PEDro and RoB2 tools were applied by two independent researchers, with any disagreement resolved through consensus.

### 2.4. Data Extraction

Data from each individual study were collected for every variable presented in Table 1, including: (1) the first author, year and country; (2) the number and characteristics of participants and the sports modality; (3) the participants’ daily caffeine intake; (4) the menstrual cycle phase and the presence of women using oral contraceptives; (5) the caffeine administration form, timing and dosage; (6) the state of fatigue when the athletes were tested (rest/fatigue condition); (7) the main performance outcomes.

For item 5, if the experiment involved different conditions besides isolated caffeine (e.g., mixing sodium phosphate with caffeine [26]), we only included the results of the isolated caffeine condition [26,27]. Regarding item 6, we identified three possible conditions: (a) fatigue: tests developed after a fatigue-inducing protocol, match or strenuous effort that would cause fatigue to the participant; (b) match or simulated match: efforts developed during a regular match situation with official rules (in some cases the match duration was modified); (c) rest: tests developed without previous fatigue. These conditions were analyzed separately, given that the effects of caffeine might be different in rested and fatigued states. In a rested condition, the aim of evaluating caffeine intake would be to assess its effect on the performance of a specific task (e.g., jump performance). However, in the fatigued condition, the main aim would be to evaluate the effect of caffeine intake in minimizing the performance decline associated with fatigue by modulating the fatigue itself or its perception (e.g., jump performance after a soccer match or at half-time). Finally, for the match variables, the main aim would be to evaluate the effects of caffeine ingestion during real match situations (accelerations, decelerations, etc.) that are influenced by both physical and cognitive factors. For item 7, the main outcomes selected were team-sport performance variables such as: jump performance; single sprint and RSA performance; agility tests; maximal voluntary isometric-, concentric- and eccentric-force tests; muscular endurance; anaerobic power (Wingate test); specific task performance (e.g., throwing a ball of the specific sport); specific match variables (body impacts, sprint speed, total sprint distance, accelerations and decelerations). We also considered the rating of perceived exertion (RPE) and fatigue indexes, as they are reliable proxies for physical performance despite not being direct outcomes of athletic performance.

### 2.5. Meta-Analyses

For the meta-analyses, we collected mean and error measures or effect sizes with confidence intervals. When these were not provided or when mean and error measures were only presented in figures, we contacted the corresponding authors [27,28,29,30,31,32,33,34] to obtain specific information (all authors replied).

Five studies included both males and females and analyzed them together, presenting pooled data. We collected mean and error values only for the female group after contacting the corresponding authors of two of the studies [14,35]. We did not include two studies [36,37] that presented both sexes because the corresponding authors confirmed that they involved the same participants and tests that were presented in the following two studies that were included in the meta-analysis [32,38]. We did not contact the authors of one study that analyzed males and females together [13] because the measured main outcomes of the study were not of interest for the present review and meta-analysis.

### 2.6. Statistical Analyses

After calculating the standardized mean difference as the individual effect size of each study for each relevant variable, the results were pooled using the DerSimonian–Laird method in a random-effects meta-analysis [39]. A minimum of three studies was required in order to perform the meta-analyses. When calculating the standardized mean difference between conditions within each meta-analysis, the data were set to indicate that a positive value always represented a difference in performance favoring the caffeine condition.

A sensitivity analysis was performed excluding those studies that administered less than 2 mg of caffeine per kg of body mass [33,40,41] as this has been suggested to be the minimum effective ergogenic dose [17].

We tested the heterogeneity using the I^2^ statistic [42]. This statistic describes the variance between studies as a proportion of the total variance. A value of 25–50% indicates low heterogeneity, between 50–75% indicates moderate heterogeneity and >75% indicates high heterogeneity.

## 3. Results

### 3.1. Main Search

The literature search provided a total of 588 studies, with 4 additional studies found through cross-referencing. A total of 54 full-text articles were read, and 18 met the inclusion criteria and were included in the systematic review. Figure 1 presents the PRISMA flow chart and the reasons for excluding articles from the final sample of selected studies.

### 3.2. Quality Assessment and Risk of Bias

The individual PEDro quality scores ranged from 8 to 10, being excellent in 15 studies and good in 3 studies (Appendix A). Three crossover trials did not meet the requirements related to therapist and assessor blinding [33,35,40], and the study performed by Fernandez-Campos et al. [41] did not include drop-outs in the analysis.

Regarding the RoB 2 tool results, 14 studies showed a low risk of bias in all domains, 4 studies demonstrated some concerns for domain 2 “bias due to deviations from intended interventions” and 3 studies showed some concerns for domain 4 “bias in measurement of the outcome” (Appendix A). Additionally, all studies had a low risk of bias in the specific domain for crossover designs “bias arising from the period effect and carryover effect” (domain S). Thus, the overall biases were low for 14 studies, with some concern in the other studies.

### 3.3. Description of Participants and Studies

Six studies were performed in Spain. Two studies were developed in the United States of America, New Zealand and Taiwan. Australia, Iran, Costa Rica, Serbia, Turkey and Singapore each provided one study. The origin of each individual study is presented in the first column of Table 1.

The main characteristics of each study are presented in Table 1. The 18 studies included provided a total of 240 young adult female TSA (the mean age for all studies was in the range of 18 to 26 years). Of this sample, 50 participants were basketball players [14,35,38,43], 40 were volleyball players [30,33,41], 37 were soccer players [29,40], 32 were rugby players [28,31] and 15 were handball players [32]. The rest of the studies used a sample of mixed TSA including basketball, volleyball, handball, soccer, rugby, softball, hockey and netball players [26,27,34,44,45,46].

### 3.4. Caffeine Supplementation and Doses

Caffeine doses ranged from 1.3 mg/kg to 6 mg/kg, mainly ingested through: capsules: nine studies [14,26,27,32,34,38,43,44,45], powders: five studies [28,29,30,31,35], energy drinks: two studies [40,41], power bars: one study [33] or coffee [46].

All studies supplied the caffeine 60 min before the experiments, except for Mahdavi et al. [43], who provided the caffeine supplementation 70 min before, Fernandez-Campos et al. [41] who provided it 30 min before and Pfeifer et al. [33] who specified that the dose “was administered immediately prior to and during the competition”.

Regarding caffeine withdrawal as part of the standardization procedures, although different instructions to volunteers were found among the studies (Specified in Table 1), most studies required participants to abstain from all dietary sources of caffeine for 48 h before the trials.

Finally, regarding the days that passed between the placebo and caffeine conditions, five studies performed washout periods of 48 to 96 h, with most studies performing one-week washout periods (10 studies). Three studies performed even longer washouts (13 to 21 days). Individual information for each study is provided in Table 1.

### 3.5. Menstrual Cycle

In the first four studies that were published between 2011 and 2014 [27,28,29,40], the phase of the menstrual cycle and the use of oral contraceptives was not reported. From 2015, some studies reported the menstrual cycle phase while others were even more strict and performed the evaluations when participants were in a specific phase. For example, Buck et al. [26] standardized the assessments with the protocol of starting in the first three days after the last menstruation (follicular phase), Chen et al. [34] instructed athletes to participate during their early follicular phase and Puente et al. [14], Stojanovic et al. [38] and Karayigit et al. [46] completed their assessments during the luteal phase. Specific phases and the use of oral contraceptives are specified in Table 1.

### 3.6. Rested, Match and Fatigued Conditions

For the rested and match conditions, enough studies were included to perform a meta-analysis, as presented below. For the fatigued condition, meta-analyses of RPE, agility, RSA and maximal voluntary isometric contraction (MVIC) were performed. We could not perform a meta-analysis for the other fitness tests in a fatigued condition due to the heterogeneity of the performed tests. Nonetheless, the last column of Table 1 presents three different symbols reflecting the effectiveness of caffeine supplementation in each individual study, with an up-arrow representing a positive effect. Of the 18 included studies, 10 performed tests including a fatigued state. Of these 10 studies, a total of 46 variables were analyzed after participants were already fatigued, finding a positive effect of caffeine for only 8 variables, as specified in the last column of Table 1.

### 3.7. Meta-Analysis Results

#### 3.7.1. Simulated Match Body Impacts

Four studies evaluated body impacts during a match, including studies of volleyball [30], basketball [14], handball [32] and rugby players [31]. The meta-analysis including total body impacts is presented in Figure 2A and shows that caffeine did improve intensity during a match (standardized mean difference (SMD): 0.488; 95% CI: 0.050, 0.927). Heterogeneity among the studies was low (I^2^ 49%, *p* = 0.117).

#### 3.7.2. Specific Sport Drills

Four studies analyzed a specific sport movement (Figure 2B). On the one hand, Perez-Lopez et al. [30] analyzed the speed of a volleyball ball in a jumping spike, while Muñoz et al. analyzed the speed of a 9 m handball throw against a goalkeeper. On the other hand, Puente et al. [14] and Tan et al. [35] measured basketball throw performance. Caffeine improved performance on overall specific sport drills (SMD: 0.384; 95% CI: 0.077, 0.691). Heterogeneity among the studies was low (I^2^ 0, *p* = 0.699). A subgroup meta-analysis was performed, showing that caffeine improved ball speed (SMD: 0.440; 95% CI: 0.098; I^2^ 0% *p* = 0.903) but did not improve effectiveness during basketball free throws (SMD: 0.150, 95% CI: −0.549, 0.849; I^2^ 0% *p* = 0.347).

#### 3.7.3. Jump Performance

Seven studies evaluated jump performance using either countermovement jumps (CMJ), Abalakov jumps (ABA) or squat jumps (SJ). One study did not describe the jump performed, calling it a vertical jump [33]. As shown in Figure 2C, caffeine showed a positive effect on CMJ performance (SMD: 0.208, 95% CI: 0.079, 0.338; I^2^ 0% *p* = 0.989). The sensitivity analysis excluding the Fernandez-Campos study due to the supplied dosage of caffeine (<2 mg/kg) revealed similar results (SMD: 0.217, 95% CI: 0.085, 0.348) with a low heterogeneity (I^2^ 0% *p* = 0.994).

Another meta-analysis was performed including three studies that measured SJ with the intake of caffeine showing no improvement in SJ performance (Figure 2D: SMD: 0.241, 95% CI: −0.189, 0.671; I^2^ 0% *p* = 0.870). The sensitivity analysis excluding the Fernandez-Campos study revealed similar results (SMD: 0.345, 95% CI: −0.237, 0.928; I^2^ 0% *p* = 0.926).

#### 3.7.4. Agility

From the six studies that evaluated agility in a rested state, three used the *t*-test [27,30,40], while one study used a modified version of the t-test [32], one study used the change-of-direction and acceleration test (CODAT) [14] and one study used the lane agility drill [38]. Figure 3A presents the performed meta-analysis including all the agility tests, showing that caffeine did not improve agility (SMD: 0.144, 95% CI: −0.127, 0.416; I^2^ 0% *p* = 0.939).

The sensitivity analysis excluding the Astorino et al. study showed similar results (SMD: 0.166, 95% CI: −0.128, 0.459; I^2^ 0% *p* = 0.892).

#### 3.7.5. Handgrip

Figure 3B shows the effects of caffeine on handgrip strength, which was measured in three studies, with Muñoz [32] reporting the mean strength of both hands in handball players, while Perez-Lopez [30] and Fernandez-Campos [41] reported separated values for the left and right hands of volleyball players (the right hand was selected for the present meta-analysis as it is usually the dominant hand). The meta-analysis showed that caffeine had a positive effect on handgrip performance (SMD: 0.395, 95% CI: 0.126, 0.665). A low heterogeneity was found (I^2^ 0% *p* = 0.476). A sensitivity analysis excluding the Fernandez-Campos study did not change the main effect of caffeine (SMD: 0.467, 95% CI: 0.047, 0.887) or the low heterogeneity (I^2^ 25% *p* = 0.238).

#### 3.7.6. Single Sprint Performance

Single sprint performance was evaluated in five studies, from which two performed a single sprint [32,38] and three performed an RSA test, with the first sprint selected for the present meta-analysis [26,28,29]. A subgroup meta-analysis was performed for studies that used a single sprint on the one hand and for studies that used the first sprint of an RSA test on the other hand. All the studies performed a 30 m sprint except for that of Stojanović and colleagues [38] who used a 20 m sprint. As presented in Figure 4A, caffeine showed no effect on single sprint performance (SMD: 0.225, 95% CI: −0.022, 0.472; I^2^ 0% *p* = 0.685). Nonetheless, when dividing studies into two groups according to the type of measurement performed (single sprint or first sprint of an RSA test), we found that those studies that performed a single sprint reported a performance improvement (SMD: 0.347, 95% CI: 0.038, 0.656; I^2^ 0% *p* = 0.903), while in those studies that performed a RSA test, caffeine did not improve the performance of the first sprint (SMD: 0.011, 95% CI: −0.399, 0.420; I^2^ 0% *p* = 0.737).

#### 3.7.7. RSA

Three studies included an RSA test, with Del Coso et al. [28] using a 6 × 30 m RSA test, Lara et al. [29] using a 7 × 30 m RSA test and Buck et al. [26] using a 6 × 20 m RSA test. As shown in Figure 4B, caffeine showed no effect on RSA (SMD: 0.155, 95% CI: −0.254, 0.565; I^2^ 0% *p* = 0.826).

#### 3.7.8. RPE

As shown in Figure 4C, nine studies evaluated RPE after performing an exercise protocol after caffeine supplementation, finding no effect of this supplement on RPE (SMD: 0.258, 95% CI: −0.048, 0.565; I^2^ 0.26% *p* = 0.260).

#### 3.7.9. Fatigued State

Six studies performed the assessments after applying a specific fatigue protocol or after a match. A meta-analysis could only be performed for the agility tests, as three studies evaluated agility after performing several all-out sprint tests [27,40] or after a match [33], with the pooled results showing no positive effects on agility after fatiguing the participants, as shown in Figure 5 (SMD: 0.069, 95% CI: −0.400, 0.538; I^2^ 0% *p* = 0.858). It should be highlighted that the caffeine doses for two of the aforementioned studies were below 2 mg/kg [33,40].

## 4. Discussion

The main findings of the present systematic review and meta-analysis suggest that oral caffeine administration before exercise has an ergogenic effect on specific team-sport skills, CMJ height, handgrip strength and total body impacts in female TSA. Nonetheless, caffeine did not show an ergogenic effect on RPE, SJ, agility, RSA or tests performed in a fatigued state.

The positive effects of caffeine supplementation on CMJ are in accordance with most of the previous systematic reviews and meta-analyses developed for team sports [8,9,10], although Ferreira and colleagues [11] did not find a positive effect of caffeine in their meta-analysis including male soccer players. Nonetheless, we also found that caffeine had no effect on SJ performance. This could partially be explained by the low number of studies (*n* = 3) evaluating the effect of caffeine on SJ performance. When considering Figure 2D, it appears that caffeine had a positive effect in the three studies, although the overall effect was not significant probably due to the low number of studies.

Muscle force and consequently CMJ could both determine specific athletic skills performance, which is of critical importance for TSA and was found to be improved by caffeine supplementation. Nonetheless, we only found a positive effect for the subgroup meta-analysis that included ball speed, which might be influenced by technique and upper-body strength/power. This would suggest that caffeine could be useful for those team sports in which upper-body strength/power is a determinant (e.g., volleyball, basketball or handball). Nevertheless, these results should be interpreted with caution, as only two studies measured ball speed, and the two studies that evaluated accuracy (through basketball free throws) found no ergogenic effects [14,35].

The improvement in CMJ performance was accompanied by an improvement in single sprint performance (when studies that only performed one sprint were included). This is in line with previous systematic reviews and meta-analyses, which found positive results in single sprints developed with TSA [8,9]. When we included the first sprint of studies that performed an RSA test the ergogenic effect of caffeine disappeared. We could therefore hypothesize that those participants who were going to complete an RSA test might not have performed the first sprint at their maximal capacity, and that caffeine supplementation does indeed have a positive effect on single sprint performance when participants are performing a single maximal-effort sprint.

Although single sprint performance is important, most team-sport athletes will need to perform several sprints during a match with short low-intensity periods between them. Consequently, several studies performed RSA tests in order to evaluate the ability of athletes to maintain sprint intensity. We found that caffeine supplementation had no effect on RSA in our meta-analysis, which disagrees with some previous systematic reviews and meta-analyses carried out with samples of men and women involved in team sports [9,10] but is in line with others [7,11]. Again, a small number of studies were included in our meta-analysis (*n* = 3), and therefore the results should be interpreted with caution, as more studies including female TSA are necessary.

The positive findings in upper-limbs isometric muscle force are in line with a recent meta-analysis developed by Grgic and Del Coso [47] focusing on the effects of caffeine on strength and power, finding that caffeine improved upper-body performance in women. This could be critical for female TSA, as an improvement in muscular endurance and strength could enable TSA to develop improved performance during a match. Along these lines, we did find improvements in total body impacts (a proxy for the players’ match intensity), which would imply higher intensities during competition. Consequently, although athletes might not be able to improve RSA under laboratory conditions, they might be more motivated during a match and be capable of improving intensity due to the ergogenic effect of caffeine. These positive findings are in line with previous meta-analyses that found improvements in the performed number of sprints during a real or simulated match after acute caffeine ingestion [9].

The lack of effect of caffeine ingestion on agility tests was surprising and contradicted results from a previous meta-analysis which included mainly male participants (two studies evaluated females out of eight studies included) [9]. These contrasting results highlight the importance of performing more research with female athletes, as the scientific community may be assuming that what works with males will work in exactly the same way with females, while we have found some differences in the current meta-analysis.

Regarding RPE, our findings are similar to those of previous meta-analyses developed for team sports [9,11] which found no effects of caffeine on RPE. These studies and ours, which are all focused on team sports, show opposite results to those found in a larger meta-analysis developed in 2005 [48] which found a 6% reduction in RPE after the ingestion of caffeine in endurance tests. In the 21 included studies (where 7 measured females), there were 13 cycling tests, 5 running tests, 2 rowing tests and 1 swimming test. It is important to notice that the aforementioned meta-analysis showed a reduction in RPE only when constant loads were applied. Team sports are characterized by numerous high-intensity efforts followed by rest periods and do not follow a constant load pattern, which could explain the lack of effect of caffeine on RPE found in the present meta-analysis. Nonetheless, this may only be partially true, as a previous study [48] also found that, although no differences were found at the end of a test to exhaustion, caffeine attenuated RPE during exercise, which could partially explain the performance improvements found in some athletes. Most of the studies included in our meta-analysis only included an RPE assessment at the end of the tests or matches, but it would be interesting for future studies to consider RPE throughout exercise. This would allow researchers to test if a reduced RPE, and therefore an increased physical performance for the same intensity, is found during a match or a laboratory test.

Along the same lines, it would be interesting to evaluate the effects of caffeine in fatigued conditions, as many of the presented studies in the current meta-analysis were developed in laboratory settings and included participants in a rested state who performed the test (agility, jumps, etc.) 60 min after the ingestion of caffeine. Nonetheless, given that TSA are usually exposed to fatiguing efforts, it would be interesting to develop more studies in fatigued conditions, as caffeine presents the ability to cross the blood–brain barrier and block the adenosine receptors in the brain, mitigating the negative effects of fatigue. Very few studies have evaluated the effects of caffeine after applying a fatigue protocol (*n* = 3, 2 and 2 for agility, RSA and MVIC, respectively). The only meta-analysis performed showed a lack of effect of caffeine on agility performance when participants were fatigued. Nonetheless, caffeine doses for two [33,40] of the three studies included were below 2 mg/kg, and therefore further studies are required to evaluate the effects of higher doses in fatigued female TSA. Although we could not perform a further meta-analysis including the fatigued state, the results from Table 1 suggest a lack of effect as the last column shows that out of the 46 variables that were registered in fatigued conditions, only 8 improved after caffeine ingestion.

In order to improve research in this topic, we would encourage future studies to report the menstrual cycle phase of participants when performing the experiments and to perform both placebo and supplementation trials during the same menstrual cycle phase, in order to reduce the possible effect of the menstrual cycle phase. The use of oral contraceptives should also be registered. Experiments should test both rested and fatigued conditions, register individual responses to caffeine which were not reported in most of the studies included in the present meta-analysis and evaluate if in “responders” lower dosages have the same effect or if increasing the dose in “non-responders” has a positive effect. This is because previous articles have identified substantial inter-individual variations following caffeine ingestion in sport [49]. These differences seem to be mediated by genetic variations, and the characterization of the athlete’s genetic profile could potentially help in individualizing the caffeine dose accurately to optimize its effect on physical performance [49].

Finally, it is worth mentioning that five of the included studies used energy drinks that, in addition to caffeine, contained other substances that could also have an ergogenic effect, such as sugar, glucuronolactone and taurine. Nonetheless, three [28,29,30] of these studies specified that the placebo drink and the energy drink were exactly the same drinks with the only difference being the caffeine content (the placebo had 0 mg/kg). The two remaining studies were those developed by Astorino et al. [40] and by Fernandez-Campos et al. [41]. Astorino et al. [40] used an energy drink containing both taurine and glucuronolactone. The taurine content was 1 g, which is far from the 6 g suggested to have an ergogenic effect [50,51]. In the case of glucuronolactone, as stated by Campos-Perez in the book *Sports and Energy drinks*: “because of the few investigations on the isolate glucuronolactone in humans, there is no evidence to support the idea of adding this compound to energy drinks to improve physical and sport performance, not even as a complement to the action of taurine and/or caffeine” [52]. Therefore, the only added ingredient that could make a difference in the included studies and influence performance was sugar, with the Astorino et al. [40] study showing a difference of 7 g (energy drink 27 g vs. placebo 20 g of sugar) and the Fernandez-Campos et al. [41] study showing a difference of 31 g (energy drink 31 g vs. placebo 0 g of sugar). Nonetheless, both studies were included in the sensitivity analyses, and we consequently repeated the meta-analyses without including them, finding similar results. Therefore, the possible effect that other ergogenic substances might have that could enhance the findings attributed to caffeine were controlled for in the present meta-analysis.

Although the present meta-analysis presents several strengths, such as the focus on an athlete population with limited previous scientific evidence (female TSA), the effort to contact the corresponding authors to obtain specific data for this group and the inclusion of the updated 2021 PRISMA guidelines, it is not without limitations, the main one being the low number of studies included in some of the meta-analyses (*n* = 3).

## 5. Conclusions

Although caffeine is generally considered as one of the most useful supplements used to increase athletic performance, the results of the present meta-analysis suggest that more research is needed in female TSA. Female TSA obtained benefits from caffein supplementation as it was shown to improve upper-body strength and sport-specific tasks related to upper-body strength (ball speed), in addition to CMJ, single sprint performance and body impacts during a match (match intensity).

## Figures and Tables

**Figure 1 nutrients-13-03663-f001:**
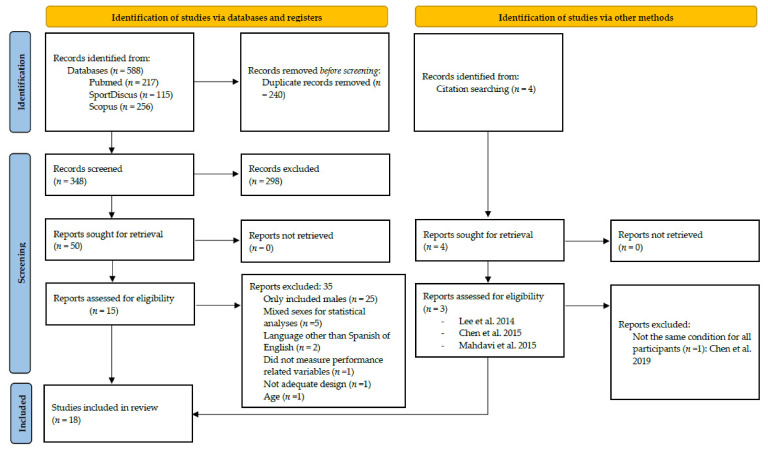
PRISMA flow-chart diagram.

**Figure 2 nutrients-13-03663-f002:**
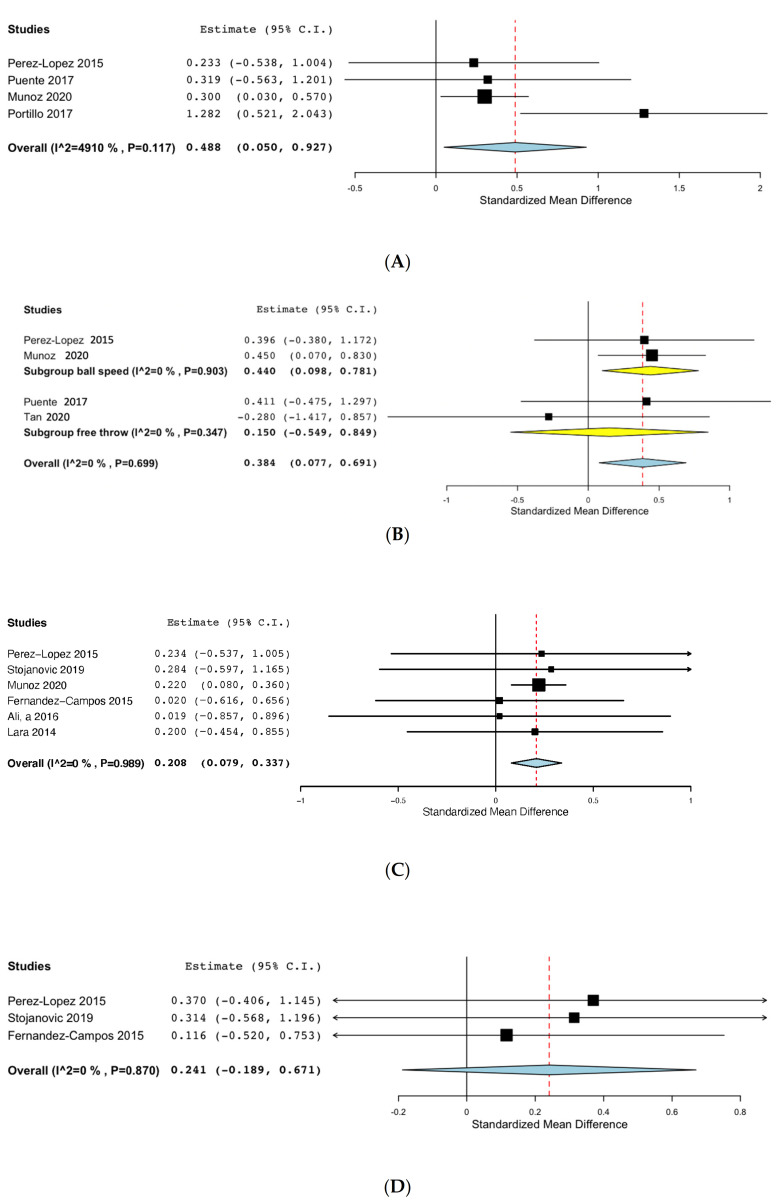
(**A**): Effects of caffeine on body impacts during a simulated match. (**B**): Effects of caffeine on specific skills. (**C**): Effects of caffeine on countermovement jump. (**D**): Effects of caffeine on squat jump.

**Figure 3 nutrients-13-03663-f003:**
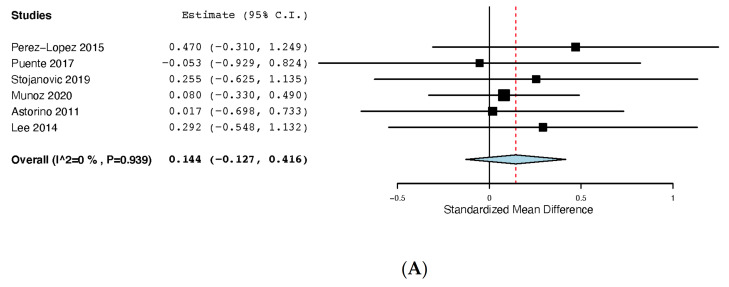
(**A**): Effects of caffeine on agility. (**B**): Effects of caffeine on handgrip strength.

**Figure 4 nutrients-13-03663-f004:**
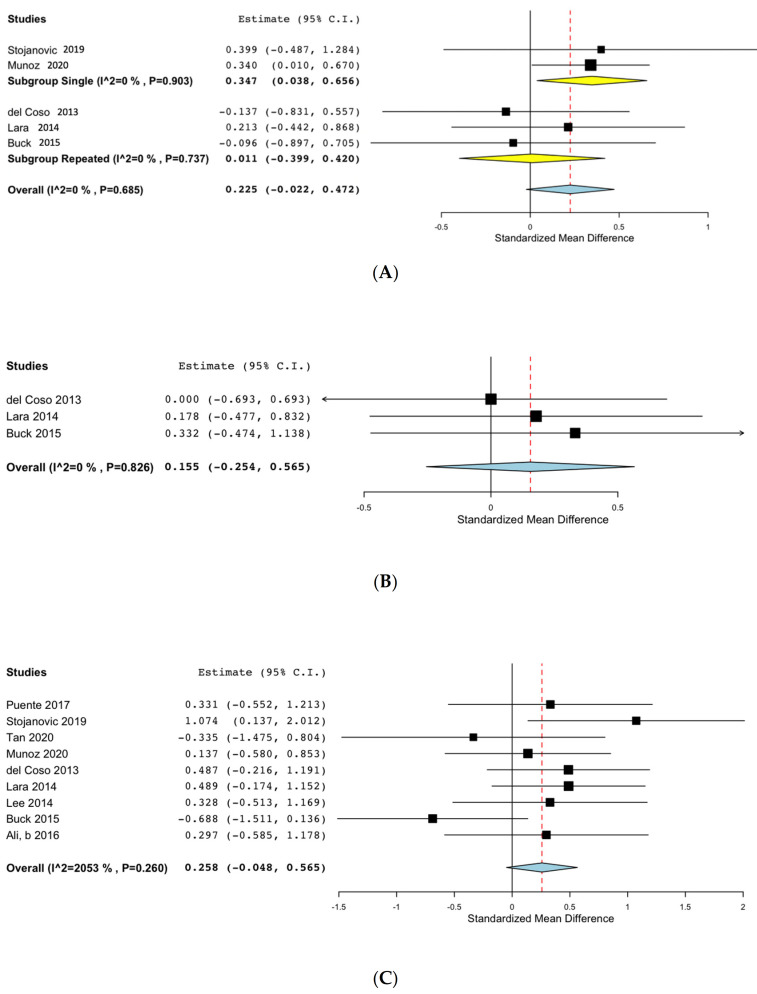
(**A**): Effects of caffeine on single sprint performance. (**B**): Effects of caffeine on repeated sprint ability performance. (**C**): Effects of caffeine on rate of perceived exertion.

**Figure 5 nutrients-13-03663-f005:**
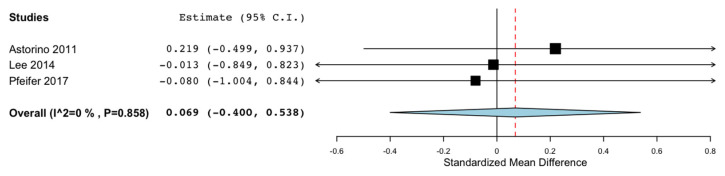
Effects of caffeine on agility after a fatigue protocol.

**Table 1 nutrients-13-03663-t001:** Characteristics of studies included in the systematic review.

Authors, Year, (Country) and PEDro Score	Sample Level ^+^	Caffeine Consumption or Restrictions	Menstrual Cycle and Oral Contraceptives	Timing + Intervention + Washout	Sample State	Outcomes	R
Astorino et al. 2011 (USA) PEDro: 8/10	15 NAIA soccer players (19.5 ± 1.1 years) Level: semi-professional	12/15 were caffeine consumers (dose not reported) Instructed not to ingest any caffeine 48 h before each trial	Not controlled	60 min pre-test CAF: Red bull (80 mg: 1.3 mg/kg) PLA: Canada dry ginger ale Washout: 72–96 h	Rest	Agility *t*-test: Set 1/3 of 8 reps.	⇄
					Fatigue	Agility *t*-test: Sets 2 and 3/3 of 8 reps	⇄
						RPE	⇄
Del Coso et al. 2013 (Spain) PEDro: 10/10	16 rugby sevens National Team (23 ± 2 years)Level: elite	Light caffeine consumers: <60 mg/day Encouraged to abstain from all dietary sources of caffeine for 48 h before	Not controlled	60 min pre-test CAF: Powder caffeine-energy drink 3 mg/kg (Fure^®^) PLA: Powder drink 0 mg/kg Washout: 72 h	Rest	6 × 30 m sprint test	⇄
					Match	Distance covered walking	⇄
						Distance covered jogging	⇄
						Distance covered cruising	↑
						Distance covered striding	↑
						Distance covered high intensity running	↑
						Distance covered sprinting	↑
						Match: RPE	⇄
					Fatigue	15 s maximal CMJs: total power	↑
Lee et al. 2014 (Taiwan) PEDro: 10/10	11 Division I collegiate team-sport athletes (Basketball or Volleyball) (21.3 ± 1.2 years) Level: semi-professional	Light caffeine consumers: 50–100 mg/day	Not controlled	60 min pre-test CAF: 6 mg/kg capsulesPLA: Cellulose capsules Washout: at least 1 week	Rest	Agility *t*-test	⇄
					Fatigue	Cycle-ergometer repeated sprint peak power	⇄
						Cycle-ergometer repeated sprint mean power	⇄
						Cycle-ergometer repeated sprint total work	⇄
						Cycle-ergometer repeated sprint decrement	⇄
						Agility *t*-test	⇄
						Blood lactate	↓
						RPE	⇄
Lara et al. 2014 (Spain) PEDro: 10/10	18 soccer players (21 ± 2 years) Level: not reported	Light caffeine consumers: not more than one cup of coffee or energy drink per day Encouraged to abstain from all dietary sources of caffeine for 48 h before	Not controlled	60 min pre-test CAF: Powder caffeine-energy drink 3 mg/kg (Fure^®^) PLA: Powder drink 0 mg/kg Washout: 1 week	Rest	7 × 30 m sprint average speed	↑
						7 × 30 m sprint maximal speed	↑
						CMJ height	↑
						CMJ Power	⇄
					Match	Total distance covered	↑
						Time standing	↑
						Time walking	⇄
						Time running (3.1–8 km/h)	↑
						Time running (8.1–13 km/h)	↑
						Time running (13.1–18 km/h)	⇄
						Time running (>18 km/h)	↑
						Number of sprint bouts	↑
						Maximal speed	⇄
						RPE	⇄
Buck et al. 2015 (Australia) PEDro: 10/10	12 amateur team-sports (netball, basketball and soccer) (25.5 ± 1.9 years) Level: amateur	Caffeine consumption not reported Participants were advised to abstain from consuming CAF for 48 h prior to each trial	3 days post (follicular phase) menstruation 9 were taking Levlen ED for birth control 3 took no oral contraceptives	60 min pre-test CAF: Capsule (6 mg/kg BM) PLA: Capsule (1 g glucose) Washout: ≈21 days	Rest	6 × 20 m sprint before PSM	⇄
						Best 6 × 20 m sprint before PSM	⇄
						Total 6 × 20 m sprint time before PSM	⇄
					Fatigue	6 × 20 m sprint half-time PSM	⇄
						6 × 20 m sprint after PSM	⇄
						Best 6 × 20 m sprint half-time PSM	⇄
						Best 6 × 20 m sprint after PSM	⇄
						Total 6 × 20 m sprint time half-time PSM	⇄
						Total 6 × 20 m sprint time after PSM	⇄
						RPE during and after PSM	⇄
						Blood lactate during and after PSM	⇄
Chen et al. 2015 (Taiwan) PEDro: 10/10	10 elite collegiate athletes (tennis, soccer, basketball) (19.9 ± 0.9 years) Level: semi-professional	No regular caffeine consumption < 200 mg/week	Instructed to participate during their early follicular phase and avoid taking contraception	60 min pre-test CAF: Capsule 6 mg/kg PLA: Diet flour in capsule Washout: 1 week	Rest	MVIC	↑
						Isometric fatigue protocol	↑
					Fatigue	Fatigued MVIC	↑
						Fatigue index	↑
						Blood lactate	↓
Mahdavi et al. 2015 (Iran) PEDro: 10/10	24 basketball players (24.2 ± 2.6 years) Level: not reported	116.8 ± 26.7 mg/day	Not controlled	70 min pre-test CAF: Capsules 5 mg/kg PLA: Capsules with dextrose Washout: 1 week	Rest	30 s WT: Peak power	⇄
						30 s WT: Mean power	⇄
						30 s WT: End power	⇄
						30 s WT: Power drop	⇄
						30 s WT: Fatigue index	⇄
						30 s WT: Lactate	↑
Fernandez-Campos et al. 2015 (Costa Rica) PEDro: 9/10	19 volleyball players from the elite league of Costa Rica (22.3 ± 4.9 years) Level: elite	Not reported	Not controlled	30 min pre-test CAF: Energy drink 6 ml/kg with 73 mg of CAF in 273 mL. (1.7 mg/kg) PLA: flavored drink Washout: 1 week	Rest	Right handgrip strength	↑
						Left handgrip strength	⇄
						CMJ height	⇄
						SJ height	⇄
						WT peak power	⇄
						WT mean power	⇄
						WT fatigue index	⇄
Perez-Lopez et al. 2015 (Spain) PEDro: 10/10	13 volleyball players from the second division of the Spanish league (25.2 ± 4.8) Level: semi-professional	On the day of the trial participants were encouraged to refrain from all dietary sources of caffeine	4 during follicular phase9 during luteal phase	60 min pre-test CAF: Powder energy drink (Fure^®^) 3 mg/kg PLA: Powder with 0 mg/kg of CAF Washout: 1 week	Rest	Handgrip	↑
						Spike jump height and peak power	↑
						Block jump height and peak power	↑
						Squat jump height and peak power	↑
						CMJ height and peak power	↑
						Agility t-test	↑
						Standing spike ball velocity	↑
						Jumping spike ball velocity	↑
					Match	Body accelerations	↑
						Positive game actions	↑
						Neutral game actions	⇄
						Negative game actions	↑
						Body impacts 0–1 g	↑
						Body impacts 1.1–2 g	↑
						Body impacts 2.1–3 g	↑
						Body impacts 3.1–4 g	⇄
						Body impacts 4.1–5 g	↑
						Body impacts 5.1–6 g	↑
Ali et al. 2016a (New Zealand) PEDro: 10/10	10 healthy team sport players (soccer, hockey and netball) (24 ± 4 years) Level: amateur and elite	Self-reported daily caffeine intake varied from 0 to 300 mg/day	All participants were taking a monophasic oral contraceptive (Monofeme, Microgynon, Levlen ED or Nordette)	60 min pre-test CAF: Capsules 6 mg/kg PLA: Capsules with artificial sweetener Washout: 13–17 days	Rest	Knee flexor ecc. PT pre-PSM	⇄
						Knee extensor ecc. PT pre-PSM	⇄
						Knee flexor ecc. Power pre-PSM	⇄
						Knee extensor ecc. Power pre-PSM	⇄
						Isometric knee flexor pre-PSM	⇄
						Isometric knee extensor pre-PSM	⇄
						CMJ height and power pre-PSM	⇄
					Fatigue	Knee flexor ecc. PT mid-PSM	↑
						Knee flexor ecc. PT post-PSM	⇄
						Knee flexor ecc. PT 12 h-post-PSM	↑
						Knee extensor ecc. PT mid-PSM	⇄
						Knee extensor ecc. PT post-PSM	⇄
						Knee extensor ecc. PT 12 h-post-PSM	⇄
						Knee flexor ecc. Power mid-PSM	↑
						Knee flexor ecc. Power post-PSM	⇄
						Knee flexor ecc. Power 12 h-post-PSM	↑
						Knee extensor ecc. Power mid-PSM	↑
						Knee extensor ecc. Power post-PSM	⇄
						Knee extensor ecc. Power 12 h-post-PSM	⇄
						Isometric knee flexor mid-PSM	⇄
						Isometric knee flexor post-PSM	⇄
						Isometric knee flexor 12 h post-PSM	⇄
						Isometric knee extensor mid-PSM	⇄
						Isometric knee extensor post-PSM	⇄
						Isometric knee extensor 12 h post-PSM	⇄
						CMJ height and power post-PSM	⇄
						CMJ height and power 12 h post-PSM	⇄
Ali et al. 2016b (New Zealand) PEDro: 10/10	10 healthy team sport players (soccer, hockey and netball) (24 ± 4 years) Level: amateur and elite	Self-reported daily caffeine intake varied from 0 to 300 mg/day	All participants were taking a monophasic oralcontraceptive	60 min pre-test CAF: Capsules 6 mg/kg PLA: Capsules with artificial sweetener Washout: 13–17 days	Fatigue	RPE	⇄
Portillo et al. 2017 (Spain) PEDro: 10/10	16 rugby sevens national team players (23 ± 2 years) Level: elite	Light caffeine consumers: <60 mg/day	Not controlled	60 min pre-test CAF: Powder 3 mg/kg PLA: Powder with 0 mg/kg of CAF Washout: 72 h	Match	Body impacts 0–6 g	↑
						Body impacts 6.01–6.5 g	↑
						Body impacts 6.51–7 g	↑
						Body impacts 7.01–8 g	⇄
						Body impacts 8.01–10 g	↑
						Body impacts > 10 g	⇄
						Frequency of technical action	⇄
						Ratings of skill performance	⇄
Puente et al. 2017 (Spain) PEDro: 10/10	10 professional basketaball players (27.9 ± 6.1 years) Level: semi-professional and elite	Light caffeine consumers < 100 mg/day Encouraged to abstain from CAF ingestion during the study	All participants were tested during their luteal phase	60 min pre-test CAF: Capsule 3 mg/kgPLA: Capsule 0 mg/kg of CAF Washout: 1 week	Rest	Abalakov jump	NA
						CODAT	
						Free throws	
						CODAT with ball	
					Match	Body impacts 0–0.99 g	
						Body impacts 1–1.99 g	
						Body impacts 2–2.99 g	
						Body impacts 3–3.99 g	
						Body impacts 4–4.99 g	
						Body impacts >5 g	
						RPE	
Pfeifer et al. 2017 (USA) PEDro: 8/10	8 volleyball NAIA volleyball (18–22 years)Level: semi-professional	CAF consumption was not restricted	Not controlled	Prior to and during the competition CAF: PowerBar^®^ PowerGel^®^ 50 mg of caffeine. Averaged 1.39 mg/kg PLA: Non-nutritive gel Washout: ≈1 week	Fatigue	Vertical jump with a two-step approach	⇄
						Three cone drill agility	⇄
						6 × 30 m sprint	⇄
Stojanovic et al. 2019 (Serbia) PEDro: 10/10	10 professional basketball players (20.2 ± 3.9 years) Level: elite	Light caffeine consumers: <100 mg/day	Completed testing in the luteal phase of their menstrual cycle Use of oral contraceptives not reported by authors	60 min pre-test CAF: Capsule (3 mg/kg BM) PLA: Capsule (Dextrose) Washout: 1 week	Rest	CMJ height	⇄
						SJ height	⇄
						ABA height	⇄
						Lane agility	⇄
						5 m sprint	⇄
						10 m sprint	↑
						20 m sprint	↑
						5 m sprint-dibbling	⇄
						10 m sprint-dibbling	⇄
						20 m sprint-dibbling	⇄
						RSP: Suicide run	⇄
						RPE	↑
Tan et al. 2020 (Singapore) PEDro: 8/10	6 basketball players Level: semi-professional	Less than 200 mg caffeine per day	Not controlled	60 min pre-test CAF: Powders (6 mg/kg BM) PLA: Powders (Maltodextrine) Washout: 72 h	Fatigue	Free throws	NA
						RPE	
Muñoz et al. 2020 (Spain) PEDro: 10/10	15 elite handball players (22.6 ± 3.6 years) Level: elite	Light caffeine consumers: 50 ± 30 mg/day	10 during follicular phase 5 during luteal phase Use of oral contraceptives not reported by authors	60 min pre-test CAF: Capsule (3 mg/kg BM) PLA: Capsule (Cellulose) Washout: 1 week	Rest	7m ball throws	↑
						9m ball throws	↑
						7m ball throws goalk.	↑
						9m ball throws goalk.	↑
						CMJ height	↑
						Handgrip	↑
						Agility: MATT	⇄
						30m sprint	↑
					Match	Accelerations frequency	↑
						Decelerations frequency	↑
						Body impacts	↑
						Total distance	⇄
						Sprint distance	⇄
						Maximal speed	⇄
						RPE	⇄
Karayigit et al. 2021 (Turkey) PEDro: 10/10	17 female team sports (rugby, handball and soccer) 23 ± 2 years) Level: elite and semi-professional	Light caffeine consumers: <25 mg/day	All sessions were performed during the luteal phase of the menstrual cycle All subjects stopped oral contraceptive consumpletion 3 months before the commencement of the study	60 min pre-test CAF: Coffee (3 mg/kg BM) (6 mg/kg BM) PLA: Decaffeinated coffee Washout: 48–72 h	Rest	3 set of repetitions to failure 40% 1 RM bench press	↑
						3 set of repetitions to failure 40% 1 RM squat	⇄

**↑**: caffeine supplementation improved performance (for variables like lactate levels, standing time or sprint times this would entail a lower lactate increase or a lower sprint and standing time); ⇄: no differences between placebo and caffeine groups; ↓: caffeine supplementation decreased performance (for variables like lactate levels, standing time or sprint times this would entail a higher lactate increase or a higher sprint and standing time). ^+^ If the participant’s category was described as recreational or amateur, the sport level was classified as amateur. If the player’s category in each study was described as collegiate or second division, the sport level was classified as semi-professional. If the player’s category in each study was elite (national team) or professional, the sport level was classified as elite. ABA: Abalakov jump; BM: body mass; CAF: caffeine group; CMJ: countermovement jump; CODAT: change-of-direction and acceleration test; Ecc.: eccentric; Goalk.: drill performed with a goalkeeper; ISOfatig: submaximal voluntary isometric fatigue protocol; Lane agility: lane agility drill; Match: match following official rules; MATT: modified version of the agility *t* test; MVIC: maximal volumetric isometric contractions; MVICpost: voluntary isometric contraction after fatigue protocol; NA: not applicable because the statistical analyses presented in the studies were performed for males and females without an individual comparison for females; NAIA: National Association of Intercollegiate athletics; Perc. Performance: perceived performance; PLA: placebo; PSM: protocol simulating the fatigue generated during a match; PT: peak torque; R: results; RPE: rate of perceived exertion; RSP: repeated sprint performance; SF: sodium phosphate; SJ: squat jump; SM: simulated match; SP: self-perceived; WT: Wingate test.

## Data Availability

Not applicable.

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
