# Peer review of "Does Acute Caffeine Supplementation Improve Physical Performance in Female Team-Sport Athletes? Evidence from a Systematic Review and Meta-Analysis"

_nutrients, 2021, doi:10.3390/nu13103663_

Round 1
Reviewer 1 Report
The work is very well presented and organized, very interesting is the summary table, which provides very useful information, it reads well even if a check of the English form would be appropriate.
I have suggestions:
- It would be appropriate to propose a mechanism (perhaps with the help of a figure) of action, perhaps highlighting the possible differences between men and women
- In the proposed studies, it should be distinguished if only caffeine or other substances were used (as in the case of energy drinks, reporting in particular whether glucuronolactone was present)
- You could do a stratification based on whether you are regular coffee consumers or not, and also propose a washout period of at least 7 days, 48h in this sense are not very effective
- Distinguish if they are professional or elite or amateur athletes
Author Response
We attach a detailed point by point response to the reviewer

Reviewer 2 Report
76. “are clear” What are you talking about?
119. AGB and GLB are the authors’ abbreviations? If yes, I suggest to delete them from the text.
121. Numbered your exclusion criteria like the inclusion criteria. Also, did you check studies which were examined the effects of caffeine via mouth rinsing?
144-145. As I suggested in row 119, delete the author's abbreviation.
181. This parameter should be added to the inclusion criteria. My sense was that you use studies only with female TSA.
198. There is a contradiction between this sentence and your exclusion criteria. In exclusion criteria, you write that studies in which were administrated doses lower than 2mg were excluded.
205-236. All this information they should be written in your methodology. As result, I am expecting to read what did you find from the review and the results of your meta-analysis.
238. 1.3 mg/kg
258. Put “table 1” immediately after its first show in the text. Or delete “table1” from paragraph 3.5.
289. Please, explain in the part of meta-analysis in methodology the abbreviation of SMD.
291. Please make sure in figures that the type of words is the same as is in the whole text.
333. Put “Figure 3B” after 3.7.5 part
361. Put “Figure 4B” after 3.7.7 RSA
363. Put “Figure 4C” after 3.7.8 RPE
378. An administrated dose below 2mg was an exclusion criterion. Wasn’t it?
384. As described in the results single sprint performance has not a clear improvement. Only, two studies show an improvement.
Author Response
We attach a point by point response to the reviewer

Round 2
Reviewer 2 Report
509. Please, it would be better to avoid the name of the energy drink. Say a famous drink or something else which hides the name of it.
Author Response
Thank you for your comment.
We have changed the name commercial name of the drink to "an energy drink".
The final text is:
Astorino et al.[40] used an energy drink containing both taurine and glucuronolactone. Regarding taurine, the content was of 1g which is far from the 6g that have been suggested to have an ergogenic effect [51,52].